# Toward Goal-Driven Neural Network Models for the Rodent Whisker-Trigeminal System

**Chengxu Zhuang**
Department of Psychology
Stanford University
Stanford, CA 94305
chengxuz@stanford.edu

**Jonas Kubilius**
Department of Brain and Cognitive Sciences
Massachusetts Institute of Technology
Cambridge, MA 02139
Brain and Cognition, KU Leuven, Belgium
qbilius@mit.edu

**Mitra Hartmann**
Departments of Biomedical Engineering
and Mechanical Engineering
Northwestern University
Evanston, IL 60208
hartmann@northwestern.edu

**Daniel Yamins**
Departments of Psychology and Computer Science
Stanford Neurosciences Institute
Stanford University
Stanford, CA 94305
yamins@stanford.edu

## Abstract

In large part, rodents "see" the world through their whiskers, a powerful tactile sense enabled by a series of brain areas that form the whisker-trigeminal system. Raw sensory data arrives in the form of mechanical input to the exquisitely sensitive, actively-controllable whisker array, and is processed through a sequence of neural circuits, eventually arriving in cortical regions that communicate with decision-making and memory areas. Although a long history of experimental studies has characterized many aspects of these processing stages, the computational operations of the whisker-trigeminal system remain largely unknown. In the present work, we take a *goal-driven* deep neural network (DNN) approach to modeling these computations. First, we construct a biophysically-realistic model of the rat whisker array. We then generate a large dataset of whisker sweeps across a wide variety of 3D objects in highly-varying poses, angles, and speeds. Next, we train DNNs from several distinct architectural families to solve a shape recognition task in this dataset. Each architectural family represents a structurally-distinct hypothesis for processing in the whisker-trigeminal system, corresponding to different ways in which spatial and temporal information can be integrated. We find that most networks perform poorly on the challenging shape recognition task, but that specific architectures from several families can achieve reasonable performance levels. Finally, we show that Representational Dissimilarity Matrices (RDMs), a tool for comparing population codes between neural systems, can separate these higher-performing networks with data of a type that could plausibly be collected in a neurophysiological or imaging experiment. Our results are a proof-of-concept that DNN models of the whisker-trigeminal system are potentially within reach.

## 1 Introduction

The sensory systems of brains do remarkable work in extracting behaviorally useful information from noisy and complex raw sense data. Vision systems process intensities from retinal photoreceptor arrays, auditory systems interpret the amplitudes and frequencies of hair-cell displacements, and somatosensory systems integrate data from direct physical interactions. [28] Although these systems

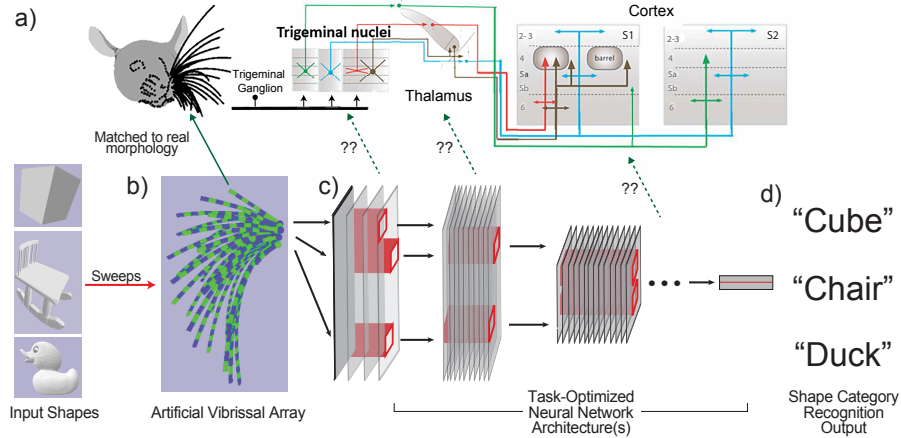

Figure 1: **Goal-Driven Approach to Modeling Barrel Cortex: a.** Rodents have highly sensitive whisker (vibrissal) arrays that provide input data about the environment. Mechanical signals from the vibrissae are relayed by primary sensory neurons of the trigeminal ganglion to the trigeminal nuclei, the original of multiple parallel pathways to S1 and S2. (Figure modified from [8].) This system is a prime target for modeling because it is likely to be richly representational, but its computational underpinnings are largely unknown. Our long-term approach to modeling the whisker-trigeminal system is *goal-driven*: using an artificial whisker-array input device built using extensive biophysical measurements (**b.**), we seek to optimize neural networks of various architectures (**c.**) to solve ethologically-relevant shape recognition tasks (**d.**), and then measure the extent to which these networks predict fine-grained response patterns in real neural recordings.

differ radically in their input modalities, total number of neurons, and specific neuronal microcircuits, they share two fundamental characteristics. First, they are hierarchical sensory cascades, albeit with extensive feedback, consisting of sequential processing stages that together produce a complex transformation of the input data. Second, they operate in inherently highly-structured spatiotemporal domains, and are generally organized in maps that reflect this structure [11].

Extensive experimental work in the rodent whisker-trigeminal system has provided insights into how these principles help rodents use their whiskers (also known as *vibrissae*) to tactually explore objects in their environment. Similar to hierarchical processing in the visual system (e.g., from V1 to V2, V4 and IT [11, 12]), processing in the somatosensory system is also known to be hierarchical[27, 17, 18]. For example, in the whisker trigeminal system, information from the whiskers is relayed from primary sensory neurons in the trigeminal ganglion to multiple trigeminal nuclei; these nuclei are the origin of several parallel pathways conveying information to the thalamus [36, 24] and then to primary and secondary somatosensory cortex (S1 and S2) [4]. However, although the rodent somatosensory system has been the subject of extensive experimental efforts[2, 26, 20, 32], there have been comparatively few attempts at computational modeling of this important sensory system.

Recent work has shown that deep neural networks (DNNs), whose architectures inherently contain hierarchy and spatial structure, can be effective models of neural processing in vision[34, 21] and audition[19]. Motivated by these successes, in this work we illustrate initial steps toward using DNNs to model rodent somatosensory systems. Our driving hypothesis is that the vibrissal-trigeminal system is optimized to use whisker-based sensor data to solve somatosensory shape-recognition tasks in complex, variable real-world environments. The underlying idea of this approach is thus to use *goal-driven* modeling (Fig 1), in which the DNN parameters — both discrete and continuous — are optimized for performance on a challenging ethologically-relevant task[35]. Insofar as shape recognition is a strong constraint on network parameters, optimized neural networks resulting from such a task may be an effective model of real trigeminal-system neural response patterns.

This idea is conceptually straightforward, but implementing it involves surmounting several challenges. Unlike vision or audition, where signals from the retina or cochlea can for many purposes be approximated by a simple structure (namely, a uniform data array representing light or sound intensities and frequencies), the equivalent mapping from stimulus (e.g. object in a scene) to sensor input in the whisker system is much less direct. Thus, a biophysically-realistic embodied model of the whisker array is a critical first component of any model of the vibrissal system. Once the sensor array is available, a second key problem is building a neural network that can accept whisker data input and use it to solve relevant tasks. Aside from the question of the neural network design itself,

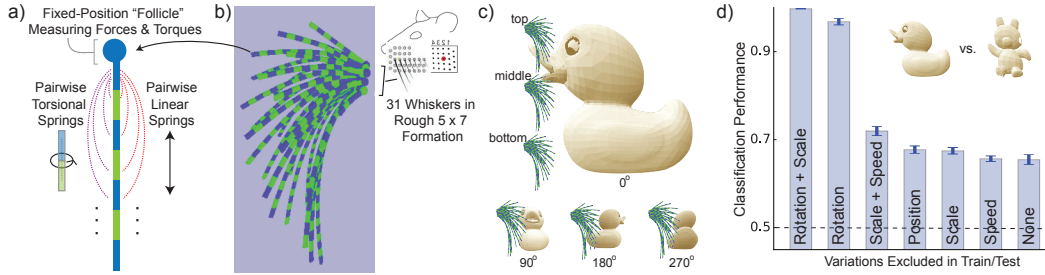

Figure 2: **Dynamic Three-Dimensional Whisker Model: a.** Each whisker element is composed of a set of cuboid links. The follicle cuboid has a fixed location, and is attached to movable cuboids making up the rest of the whisker. Motion is constrained by linear and torsional springs between each pair of cuboids. The number of cuboid links and spring equilibrium displacements are chosen to match known whisker length and curvature [31], while damping and spring stiffness parameters are chosen to ensure mechanically plausible whisker motion trajectories. **b.** We constructed a 31-whisker array, arranged in a rough 5x7 grid (with 4 missing elements) on an ellipsoid representing the rodent's mystacial pad. Whisker number and placement was matched to the known anatomy of the rat [31]. **c.** During dataset construction, the array is brought into contact with each object at three vertical heights, and four 90°-separated angles, for a total of 12 sweeps. The object's size, initial orientation angle, as well as sweep speed, vary randomly between each group of 12 sweeps. Forces and torques are recorded at the three cuboids closest to the follicle, for a total of 18 measurements per whisker at each timepoint. **d.** Basic validation of performance of binary linear classifier trained on raw sensor output to distinguish between two shapes (in this case, a duck versus a teddy bear). The classifier was trained/tested on several equal-sized datasets in which variation on one or more latent variable axes has been suppressed. "None" indicates that all variations are present. Dotted line represents chance performance (50%).

knowing what the "relevant tasks" are for training a rodent whisker system, in a way that is sufficiently concrete to be practically actionable, is a significant unknown, given the very limited amount of ethologically-relevant behavioral data on rodent sensory capacities[32, 22, 25, 1, 9]. Collecting neural data of sufficient coverage and resolution to quantitatively evaluate one or more task-optimized neural network models represents a third major challenge. In this work, we show initial steps toward the first two of these problems (sensor modeling and neural network design/training).

## 2 Modeling the Whisker Array Sensor

In order to provide our neural networks inputs similar to those of the rodent vibrissal system, we constructed a physically-realistic three-dimensional (3D) model of the rodent vibrissal array (Fig. 2). To help ensure biological realism, we used an anatomical model of the rat head and whisker array that quantifies whisker number, length, and intrinsic curvature as well as relative position and orientation on the rat's face [31]. We wanted the mechanics of each whisker to be reasonably accurate, but at the same time, also needed simulations to be fast enough to generate a large training dataset. We therefore used the Bullet [33], an open-source real-time physics engine used in many video games.

**Statics.** Individual whiskers were each modeled as chains of "cuboid" links with a square cross-section and length of 2mm. The number of links in each whisker was chosen to ensure that the total whisker length matched that of the corresponding real whisker (Fig. 2 a). The first (most proximal) link of each simulated whisker corresponded to the follicle at the whisker base, where the whisker inserts into the rodent's face. Each whisker follicle was fixed to a single location in 3D space. The links of the whisker are given first-order linear and rotational damping factors to ensure that unforced motions dissipate over time. To simplify the model, the damping factors were assumed to be the same across all links of a given whisker, but different from whisker to whisker. Each pair of links within a whisker was connected with linear and torsional first-order springs; these springs both have two parameters (equilibrium displacement and stiffness). The equilibrium displacements of each spring were chosen to ensure that the whisker's overall static shape matched the measured curvature for the corresponding real whisker. Although we did not specifically seek to match the detailed biophysics of the whisker mechanics (e.g. the fact that the stiffness of the whisker increases with the 4th power of its radius), we assumed that the stiffness of the springs spanning a given length were linearly correlated to the distance between the starting position of the spring and the base, roughly capturing the fact that the whisker is thicker and stiffer at the bottom [13].

The full simulated whisker array consisted of 31 simulated whiskers, ranging in length from 8mm to 60mm (Fig. 2b). The fixed locations of the follicles of the simulated whiskers were placed on a curved ellipsoid surface modeling the rat's mystacial pad (cheek), with the relative locations of

the follicles on this surface obtained from the morphological model [31], forming roughly a $5 \times 7$ grid-like pattern with four vacant positions.

**Dynamics.** Whisker dynamics are generated by collisions with moving three-dimensional rigid bodies, also modeled as Bullet physics objects. The motion of a simulated whisker in reaction to external forces from a collision is constrained only by the fixed spatial location of the follicle, and by the damped dynamics of the springs at each node of the whisker. However, although the spring equilibrium displacements are determined by static measurements as described above, the damping factors and spring stiffnesses cannot be fully determined from these data. If we had detailed dynamic trajectories for all whiskers during realistic motions (e.g. [29]), we would have used this data to determine these parameters, but such data are not yet available.

In the absence of empirical trajectories, we used a heuristic method to determine damping and stiffness parameters, maximizing the "mechanical plausibility" of whisker behavior. Specifically, we constructed a battery of scenarios in which forces were applied to each whisker for a fixed duration. These scenarios included pushing the whisker tip towards its base (axial loading), as well as pushing the whisker parallel or perpendicular to its intrinsic curvature (transverse loading in or out of the plane of intrinsic curvature). For each scenario and each potential setting of the unknown parameters, we simulated the whisker's recovery after the force was removed, measuring the maximum displacement between the whisker base and tip caused by the force prior to recovery ($d$), the total time to recovery ($T$), the average arc length travelled by each cuboid during recovery ($S$), and the average translational speed of each cuboid during recovery ($v$). We used metaparameter optimization [3] to automatically identify stiffness and damping parameters that simultaneously minimized the time and complexity of the recovery trajectory, while also allowing the whisker to be flexible. Specifically, we minimized the loss function $0.025S + d + 20T - 2v$, where the coefficients were set to make terms of comparable magnitude. The optimization was performed for every whisker independently, as whisker length and curvature interacts nonlinearly with its recovery dynamics.

## 3 A Large-Scale Whisker Sweep Dataset

Using the whisker array, we generated a dataset of whisker responses to a variety of objects.

**Sweep Configuration.** The dataset consists of series of simulated sweeps, mimicking one action in which the rat runs its whiskers past an object while holding its whiskers fixed (no active whisking). During each sweep, a single 3D object moves through the whisker array from front to back (rostral to caudal) at a constant speed. Each sweep lasts a total of one second, and data is sampled at 110Hz. Sweep scenarios vary both in terms of the identity of the object presented, as well as the position, angle, scale (defined as the length of longest axis), and speed at which it is presented. To simulate observed rat whisking behavior in which animals often sample an object at several vertical locations (head pitches) [14], sweeps are performed at three different heights along the vertical axis and at each of four positions around the object ($0°$, $90°$, $180°$, and $270°$ around the vertical axis), for a total of 12 sweeps per object/latent variable setting (Fig. 2c).

Latent variables settings are sampled randomly and independently on each group of sweeps, with object rotation sampled uniformly within the space of all 3D rotations, object scale sampled uniformly between 25-135mm, and sweep speed sampled randomly between 77-154mm/s. Once these variables are chosen, the object is placed at a position that is chosen uniformly in a $20 \times 8 \times 20\text{mm}^3$ volume centered in front of the whisker array at the chosen vertical height, and is moved along the ray toward the center of the whisker array at the chosen speed. The position of the object may be adjusted to avoid collisions with the fixed whisker base ellipsoid during the sweep. See supplementary information for details.

The data collected during a sweep includes, for each whisker, the forces and torques from all springs connecting to the three cuboids most proximate to the base of the whisker. This choice reflects the idea that mechanoreceptors are distributed along the entire length of the follicle at the whisker base [10]. The collected data comprises a matrix of shape $110 \times 31 \times 3 \times 2 \times 3$, with dimensions respectively corresponding to: the 110 time samples; the 31 spatially distinct whiskers; the 3 recorded cuboids; the forces and torques from each cuboid; and the three directional components of force/torque.

**Object Set.** The objects used in each sweep are chosen from a subset of the ShapeNet [6] dataset, which contains over 50,000 3D objects, each with a distinct geometry, belonging to 55 categories. Because the 55 ShapeNet categories are at a variety of levels of within-category semantic similarity, we refined the original 55 categories into a taxonomy of 117 (sub)categories that we felt had a more

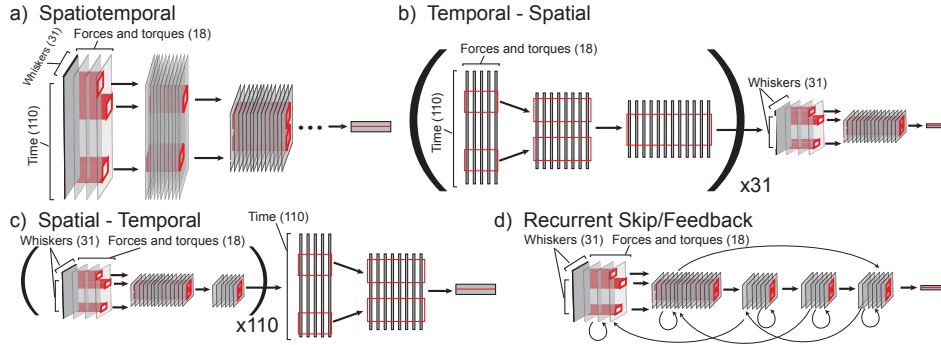

Figure 3: **Families of DNN Architectures tested: a.** "Spatiotemporal" models include spatiotemporal integration at all stages. Convolution is performed on both spatial and temporal data dimensions, followed by one or several fully connected layers. **b.** "Temporal-Spatial" networks in which temporal integration is performed separately before spatial integration. Temporal integration consists of one-dimensional convolution over the temporal dimension, separately for each whisker. In spatial integration stages, outputs from each whisker are registered to their natural two-dimensional (2D) spatial grid and spatial convolution performed. **c.** In "Spatial-Temporal" networks, spatial convolution is performed first, replicated with shared weights across time points; this is then followed by temporal convolution. **d.** Recurrent networks do not explicitly contain separate units to handle different discrete timepoints, relying instead on the states of the units to encode memory traces. These networks can have local recurrence (e.g. simple addition or more complicated motifs like LSTMs or GRUs), as well as long-range skip and feedback connections.

uniform amount of within-category shape similarity. The distribution of number of ShapeNet objects is highly non-uniform across categories, so we randomly subsampled objects from large categories. This procedure ensured that all categories contained approximately the same number of objects. Our final object set included 9,981 objects in 117 categories, ranging between 41 and 91 object exemplars per category (mean=85.3, median=91, std=10.2, see supplementary material for more details). To create the final dataset, for every object, 26 independent samples of rotation, scaling, and speed were drawn and the corresponding group of 12 sweeps created. Out of these 26 sweep groups, 24 were added to a training subset, while the remainder were reserved for testing.

**Basic Sensor Validation.** To confirm that the whisker array was minimally functional before proceeding to more complex models, we produced smaller versions of our dataset in which sweeps were sampled densely for two objects (a bear and a duck). We also produced multiple easier versions of this dataset in which variation along one or several latent variables was suppressed. We then trained binary support vector machine (SVM) classifiers to report object identity in these datasets, using only the raw sensor data as input, and testing classification accuracy on held-out sweeps (Fig. 2d). We found that with scale and object rotation variability suppressed (but with speed and position variability retained), the sensor was able to nearly perfectly identify the objects. However, with all sources of variability present, the SVM was just above chance in its performance, while combinations of variability are more challenging for the sensor than others (details can be found in supplementary information). Thus, we concluded that our virtual whisker array was basically functional, but that unprocessed sensor data cannot be used to directly read out object shape in anything but the most highly controlled circumstances. As in the case of vision, it is exactly this circumstance that calls for a deep cascade of sensory processing stages.

## 4 Computational Architectures

We trained deep neural networks (DNNs) in a variety of different architectural families (Fig. 3). These architectural families represent qualitatively different classes of hypotheses about the computations performed by the stages of processing in the vibrissal-trigeminal system. The fundamental questions explored by these hypotheses are how and where temporal and spatial information are integrated. Within each architectural family, the differences between specific parameter settings represent nuanced refinements of the larger hypothesis of that family. Parameter specifics include how many layers of each type are in the network, how many units are allocated to each layer, what kernel sizes are used at each layer, and so on. Biologically, these parameters may correspond to the number of brain regions (areas) involved, how many neurons these regions have relative to each other, and neurons' local spatiotemporal receptive field sizes [35].

**Simultaneous Spatiotemporal Integration.** In this family of networks (Fig. 3a), networks consisted of convolution layers followed by one or more fully connected layers. Convolution is performed

simultaneously on both temporal and spatial dimensions of the input (and their corresponding downstream dimensions). In other words, temporally-proximal responses from spatially-proximal whiskers are combined together simultaneously, so that neurons in each successive layers have larger receptive fields in both spatial and temporal dimensions at once. We evaluated both 2D convolution, in which the spatial dimension is indexed linearly across the list of whiskers (first by vertical columns and then by lateral row on the $5 \times 7$ grid), as well as 3D convolution in which the two dimensions of the $5 \times 7$ spatial grid are explicitly represented. Data from the three vertical sweeps of the same object were then combined to produce the final output, culminating in a standard softmax cross-entropy.

**Separate Spatial and Temporal Integration.** In these families, networks begin by integrating temporal and spatial information separately (Fig. 3b-c). One subclass of these networks are "Temporal-Spatial" (Fig. 3b), which first integrate temporal information for each individual whisker separately and then combine the information from different whiskers in higher layers. Temporal processing is implemented as 1-dimensional convolution over the temporal dimension. After several layers of temporal-only processing (the number of which is a parameter), the outputs at each whisker are then reshaped into vectors and combined into a $5 \times 7$ whisker grid. Spatial convolutions are then applied for several layers. Finally, as with the spatiotemporal network described above, features from three sweeps are concatenated into a single fully connected layer which outputs softmax logits.

Conversely, "Spatial-Temporal" networks (Fig. 3c) first use 2D convolution to integrate across whiskers for some number of layers, with shared parameters between the copies of the network for each timepoint. The temporal sequence of outputs is then combined, and several layers of 1D convolution are then applied in the temporal domain. Both Temporal-Spatial and Spatial-Temporal networks can be viewed as subclasses of 3D simultaneous spatiotemporal integration in which initial and final portions of the network have kernel size 1 in the relevant dimensions. These two network families can thus be thought of as two different strategies for allocating parameters between dimensions, i.e. different possible biological circuit structures.

**Recurrent Neural Networks with Skip and Feedback Connections.** This family of networks (Fig. 3d) does not allocate units or parameters explicitly for the temporal dimension, and instead requires temporal processing to occur via the temporal update evolution of the system. These networks are built around a core feedforward 2D spatial convolution structure, with the addition of (i) local recurrent connections, (ii) long-range feedforward skips between non-neighboring layers, and (iii) long-range feedback connections. The most basic update rule for the dynamic trajectory of such a network through (discrete) time is: $H_{t+1}^i = F_i \left( \oplus_{j \neq i} R_t^j \right) + \tau_i H_t^i$ and $R_t^i = A_i[H_t^i]$, where $R_t^i$ and $H_t^i$ are the output and hidden state of layer $i$ at time $t$ respectively, $\tau_i$ are decay constants, $\oplus$ represents concatenation across the channel dimension with appropriate resizing to align dimensions, $F_i$ is the standard neural network update function (e.g. 2-D convolution), and $A_i$ is activation function at layer $i$. The learned parameters of this type of network include the values of the parameters of $F_i$, which comprises both the feedforward and feedback weights from connections coming in to layer $i$, as well as the decay constants $\tau_i$. More sophisticated dynamics can be incorporated by replacing the simple additive rule above with a local recurrent structure such as Long Short-Term Memory (LSTM) [15] or Gated Recurrent Networks (GRUs) [7].

## 5   Results

**Model Performance:** Our strategy in identifying potential models of the whisker-trigeminal system is to explore many specific architectures within each architecture family, evaluating each specific architecture both in terms of its ability to solve the shape recognition task in our training dataset, and its efficiency (number of parameters and number of overall units). Because we evaluate networks on held-out validation data, it is not inherently unfair to compare results from networks different numbers of parameters, but for simplicity we generally evaluated models with similar numbers of parameters: exceptions are noted where they occur. As we evaluated many individual structures within each family, a list of the specific models and parameters are given in the supplementary materials.

Our results (Fig. 4) can be summarized with following conclusions:

- Many specific network choices within all families do a poor job at the task, achieving just-above-chance performance.
- However, within each family, certain specific choices of parameters lead to much better network performance. Overall, the best performance was obtained for the Temporal-Spatial model, with

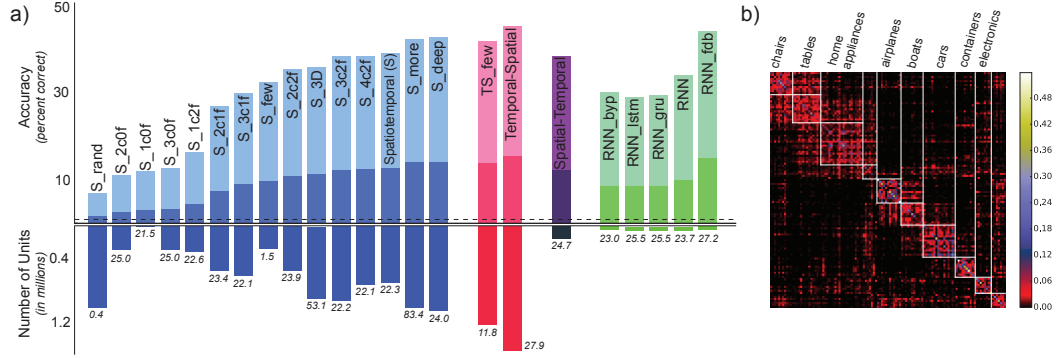

Figure 4: **Performance results. a.** Each bar in this figure represents one model. The positive $y$-axis is performance measured in percent correct (top1=dark bar, chance=0.85%, top5=light bar, chance=4.2%). The negative $y$-axis indicates the number of units in networks, in millions of units. Small italic numbers indicate number of model parameters, in millions. Model architecture family is indicated by color. "ncmf" means n convolution and m fully connected layers. Detailed definition of individual model labels can be found in supplementary material. **b.** Confusion Matrix for the highest-performing model (in the Temporal-Spatial family). The objects are regrouped using methods described in supplementary material.

> 15.2% top-1 and 44.8% top-5 accuracy. Visualizing a confusion matrix for this network (Fig. 4)b and other high-performing networks indicate that the errors they make are generally reasonable.

- Training the filters was extremely important for performance; no architecture with random filters performed above chance levels.

- Architecture depth was an important factor in performance. Architectures with fewer than four layers achieved substantially lower performance than somewhat deeper ones.

- Number of model parameters was a somewhat important factor in performance within an architectural family, but only to a point, and not between architectural families. The Temporal-Spatial architecture was able to outperform other classes while using significantly fewer parameters.

- Recurrent networks with long-range feedback were able to perform nearly as well as the Temporal-Spatial model with equivalent numbers of parameters, while using far fewer units. These long-range feedbacks appeared critical to performance, with purely local recurrent architectures (including LSTM and GRU) achieving significantly worse results.

**Model Discrimination:** The above results indicated that we had identified several high-performing networks in quite distinct architecture families. In other words, the strong performance constraint allows us to identify several specific candidate model networks for the biological system, reducing a much larger set of mostly non-performing neural networks into a "shortlist". The key biologically relevant follow-up question is then: how should we distinguish between the elements in the shortlist? That is, what reliable signatures of the differences between these architectures could be extracted from data obtainable from experiments that use today's neurophysiological tools?

To address this question, we used Representational Dissimilarity Matrix (RDM) analysis [23]. For a set of stimuli $S$, RDMs are $|S| \times |S|$-shaped correlation distance matrices taken over the feature dimensions of a representation, e.g. matrices with $ij$-th entry $RDM[i, j] = 1 - corr(F[i], F[j])$ for stimuli $i, j$ and corresponding feature output $F[i], F[j]$. The RDM characterizes the geometry of stimulus representation in a way that is independent of the individual feature dimensions. RDMs can thus be quantitatively compared between different feature representations of the same data. This procedure been useful in establishing connections between deep neural networks and the ventral visual stream, where it has been shown that the RDMs of features from different layers of neural networks trained to solve categorization tasks match RDMs computed from visual brain areas at different positions along the ventral visual hierarchy [5, 34, 21]. RDMs are readily computable from neural response pattern data samples, and are in general comparatively robust to variability due to experimental randomness (e.g. electrode/voxel sampling). RDMs for real neural populations from the rodent whisker-trigeminal system could be obtained through a conceptually simple electrophysiological recording experiment similar in spirit to those performed in macaque [34].

We obtained RDMs for several of our high-performing models, computing RDMs separately for each model layer (Fig. 5a), averaging feature vectors over different sweeps of the same object before

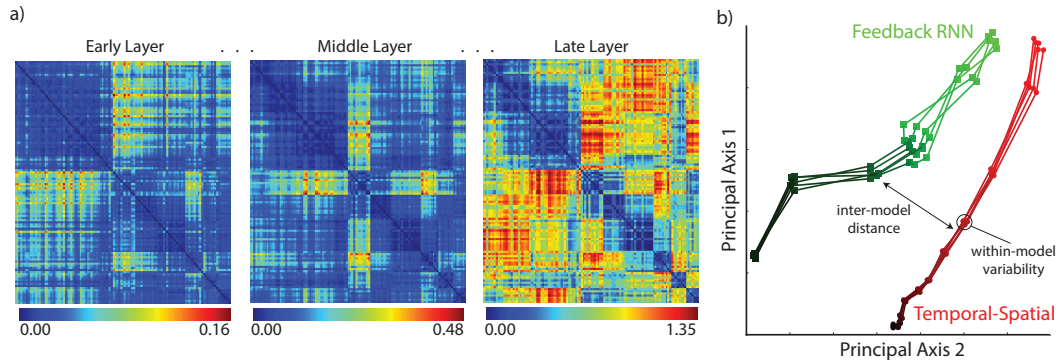

Figure 5: **Using RDMs to Discriminate Between High-Performing Models. a.** Representational Dissimilarity Matrices (RDMs) for selected layers of a high-performing network from Fig. 4a, showing early, intermediate and late model layers. Model feature vectors are averaged over classes in the dataset prior to RDM computation, and RDMs are shown using the same ordering as in Fig. 4b. **b.** Two-dimensional MDS embedding of RDMs for the feedback RNN (green squares) and Temporal-Spatial (red circles) model. Points correspond to layers, lines are drawn between adjacent layers, with darker color indicating earlier layers. Multiple lines are models trained from different initial conditions, allowing within-model noise estimate.

computing the correlations. This procedure lead to $9981 \times 9981$-sized matrices (there were 9,981 distinct object in our dataset). We then computed distances between each layer of each model in RDM space, as in (e.g.) [21]. To determine if differences in this space between models and/or layers were significant, we computed RDMs for multiple instances of each model trained with different initial conditions, and compared the between-model to within-model distances. We found that while the top layers of models partially converged (likely because they were all trained on the same task), intermediate layers diverged substantially between models, by amounts larger than either the initial-condition-induced variability within a model layer or the distance between nearby layers of the same model (Fig. 5b). This observation is important from an experimental design point of view because it shows that different model architectures differ substantially on a well-validated metric that may be experimentally feasible to measure.

## 6 Conclusion

We have introduced a model of the rodent whisker array informed by biophysical data, and used it to generate a large high-variability synthetic sweep dataset. While the raw sensor data is sufficiently powerful to separate objects at low amounts of variability, at higher variation levels deeper non-linear neural networks are required to extract object identity. We found further that while many particular network architectures, especially shallow ones, fail to solve the shape recognition task, reasonable performance levels can be obtained for specific architectures within each distinct network structural family tested. We then showed that a population-level measurement that is in principle experimentally obtainable can distinguish between these higher-performing networks. To summarize, we have shown that a goal-driven DNN approach to modeling the whisker-trigeminal system is feasible. Code for all results, including the whisker model and neural networks, is publicly available at `https://github.com/neuroailab/whisker_model`.

We emphasize that the present work is proof-of-concept rather than a model of the real nervous system. A number of critical issues must be overcome before our true goal — a full integration of computational modeling with experimental data — becomes possible. First, although our sensor model was biophysically informed, it does not include active whisking, and the mechanical signals at the whisker bases are approximate [29, 16].

An equally important problem is that the goal that we set for our network, i.e. shape discrimination between 117 human-recognizable object classes, is not directly ethologically relevant to rodents. The primary reason for this task choice was practical: ShapeNet is a readily available and high-variability source of 3D objects. If we had instead used a small, manually constructed, set of highly simplified objects that we hoped were more "rat-relevant", it is likely that our task would have been too simple to constrain neural networks at the scale of the real whisker-trigeminal system. Extrapolating from modeling of the visual system, training a deep net on 1000 image categories yields a feature basis that can readily distinguish between previously-unobserved categories [34, 5, 30]. Similarly, we suggest that the large and variable object set used here may provide a meaningful constraint on network

structure, as the specific object geometries may be less important then having a wide spectrum of such geometries. However, a key next priority is systematically building an appropriately large and variable set of objects, textures or other class boundaries that more realistically model the tasks that a rodent faces. The specific results obtained (e.g. which families are better than others, and the exact structure of learned representations) are likely to change significantly when these improvements are made.

In concert with these improvements, we plan to collect neural data in several areas within the whisker-trigeminal system, enabling us to make direct comparisons between model outputs and neural responses with metrics such as the RDM. There are few existing experimentally validated signatures of the computations in the whisker-trigeminal system. Ideally, we will validate one or a small number of the specific model architectures described above by identifying a detailed mapping of model internal layers to brain-area specific response patterns. A core experimental issue is the magnitude of real experimental noise in trigeminal-system RDMs. We will need to show that this noise does not swamp inter-model distances (as shown in Fig. 5b), enabling us to reliably identify which model(s) are better predictors of the neural data. Though real neural RDM noise cannot yet be estimated, the intermodel RDM distances that we can compute computationally will be useful for informing experimental design decisions (e.g. trial count, stimulus set size, &c).

In the longer term, we expect to use detailed encoding models of the whisker-trigeminal system as a platform for investigating issues of representation learning and sensory-based decision making in the rodent. A particularly attractive option is to go beyond fixed class discrimination problems and situate a synthetic whisker system on a mobile animal in a navigational environment where it will be faced with a variety of actively-controlled discrete and continuous estimation problems. In this context, we hope to replace our currently supervised loss function with a more naturalistic reinforcement-learning based goal. By doing this work with a rich sensory domain in rodents, we seek to leverage the sophisticated neuroscience tools available in these systems to go beyond what might be possible in other model systems.

## 7    Acknowledgement

This project has sponsored in part by hardware donation from the NVIDIA Corporation, a James S. McDonnell Foundation Award (No. 220020469) and an NSF Robust Intelligence grant (No. 1703161) to DLKY, the European Union's Horizon 2020 research and innovation programme (No. 705498) to JK, and NSF awards (IOS-0846088 and IOS-1558068) to MJZH.

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
