[Supplementary Material]

# Supplementary information

## 1 Dataset generation

### 1.1 ShapeNet category refinement and object preprocessing

Starting from the original 55 categories in ShapeNet [2], we further split them into 117 categories based on the information provided by ShapeNet. For every object in ShapeNet, beside the label from 55 categories, a tree of synsets in WordNet [8] is also provided where the object will be described in more and more detail. For example, one tree could include "artifact", "instrumentation", "transport", "vehicle", "craft", "vessel", and finally "ship". The deeper for one synset in the tree, more refined that synset will be. The label provided is usually not the deepest synset in the tree, which gives rise for refinement. Therefore, we first regrouped the objects in ShapeNet using their deepest synset in synset tree. Then we manually combined some of them as they represent the same thing, such as "liquid crystal display" and "computer monitor". In order to get a dataset with balanced category distribution, we dropped the subcategories containing less than 30 objects, which left 117 subcategories (a full list of the categories can be found in Table 4). Furthermore, with the aim of including roughly 10000 objects with balanced category distribution, we first sorted the categories by the number of objects contained and then sampled objects from smallest category to largest category sequentially. For every sampling, we would first multiply the number of objects in this category with the number of categories left ($c_l$). If the result was smaller than 10000 minus the number of objects already sampled ($n_a$), we would just take all of objects in this category. Otherwise, we would randomly sample $(10000 - n_a)/c_l$ objects from each category left. Finally, we got 9981 objects in 117 categories, with most categories containing 91 objects and smallest category containing 41 objects.

The correct collision simulation in Bullet [9] requires the object to be composed of several convex shapes. Therefore, we used V-HACD [7] to decompose every object. The decomposation is done to each object with "resolution" parameter being 500000, "maxNumVerticesPerCH" being 64, and other parameters being default.

### 1.2 Sweep simulation procedure

Two technical details about sweep simulatin would be described in this section, including rescaling of the object and the adjusting of the position to avoid

collision with fixed whisker bases.

For each object, we first computed a cuboid cover for the whole shape by getting the maximal and minimal values of points on the object in three dimensions. The longest distance in the cuboid cover would then be used as the current scale of this object. And scaling factor was computed by dividing the desired scale by the longest distance.

After rescaling, the object was moved to make the center of cuboid cover to be placed at the desired position and then the nearest point on the object to the center of whisker array was computed. The object was moved again to make the nearest point to be placed at the desired position. To avoid collision, we sampled points on the surfaces of cuboid cover and then computed the trajectories of these points. We would move the object to right to make the nearest distance from every fixed base to every trajectory larger than 4mm.

## 1.3   Basic control experiment

We generated 24000 independent sweeps for each version of control dataset, meaning 12000 for each category. For each dataset, we split the dataset equally to three parts containing 8000 sweeps overall. We then took 7000 of them as training dataset and the left as testing dataset. To reduce the over-fitting, we randomly sampled 1000 data from input data and the sampling remained the same for all control datasets. The number of units sampled has been searched to make sure that performance could not be added with more units. A LinearSVM is then trained to do the classification with a grid search of parameters. The standard deviation shown on the figure was based on the performances across three splits in one control dataset.

# 2   Model training and visualization

## 2.1   Training procedure

We searched the learning parameter space including the choices of learning algorithm, learning rates, and decay rates. Among them, training using Adagrad [3] with a initial learning rates at 0.003 and batch size of 128 gave the best result, which was then applied to the training of all networks. The learning rate would not be changed to 0.0015 until the performance on validation saturated. Ten more epochs would be run after changing the learning rate to 0.0015. The reported performances were the best performances on validation during the whole training procedure.

## 2.2   Model structures

To make the description of model structures easier, we would use conv(*filter size*, *number of filters*), pool(*filter size*, *stride*), and fc(*dimension of outputs*) to represent convolution layer, pooling layer, and fully connected layer respectively. The size of stride in convolution layers is set to be $1 \times 1$ for all networks and the

pooling used in our networks is always max-pooling. For example, conv$(3 \times 3, 384)$ represents a convolution layer with filter size of $3 \times 3$, stride of $1 \times 1$, and 384 filters.

We used Xavier initialization [5] for the weight matrix in convolution layers while the bias was all initialized to 1. The weight matrix in fully connected layer was initialized using truncated normal distribution with mean being 0 and standard deviation being 0.01, where values deviated from mean for more than two standard deviations would be discarded and redrawn. The bias in fully connected layer was initialized to 0.1. A ReLu layer would always be added after convolution or fully connected layer. And for the fully connected layers, we would use a dropout of rate 0.5 during training. [6]

**Simultaneous Spatiotemporal Integration.** This family of networks (family S) will usually have several convolution layers followed by fully connected layers. The convolution is applied to both the temporal and spatial dimension.

In Table 1, we showed the structure of all networks in this family, corresponding to the name used in the main paper. The output of previous layer will serve as the input to the next layer. And the layers before fc_combine are shared across three sweeps. For example, in model "S", conv1 to fc7 are shared across sweeps while the outputs of fc7 will be concatenated together as the input to fc_combine8. Besides, S_rand is the same model with S_more except that the weights of conv1 to fc7 there are not trained. Only fc_combine8 is trained to prove that training of weights is necessary.

Separately, we trained another network based on S_more combining 12 sweeps rather than 3 sweeps. The model shared the same structure as S_more. But fc_combine8 there would combine information from 12 sweeps, which means that the input to fc_combine8 is a vector of $1024 \times 12$. The performances of this network surpassed all other networks, with top1 being 0.20 and top5 being 0.53. This means that our network could utilize information from 12 sweeps to help finish the task.

**Separate Spatial and Temporal Integration.** The network structures for those two families are shown in Table 2. For the "Temporal-Spatial" family, temporal convolution is applied to the inputs first for each whisker using shared weights and then "spatial regroup" in Table 2 means that the outputs from previous layer for each whisker will be grouped according to the spatial position of each whisker in $5 \times 7$ grid, with vacant positions filled by zeros. For the "Spatial-Temporal" family, the input is first split into 22 vectors on temporal dimension and each vectore is further reshaped into 2D spatial grid, which means the final shape of each vector would be $5 \times 7 \times 90$. Spatial convolution networks with shared weights will be applied to each vector. After that, the "temporal concatenating" in Table 2 means that the outputs from spatial networks will first be reshaped to one dimension vector and then be concatenated in the temporal dimension to form a new input for further processing. Then temporal convolution will be applied to the new input.

**Recurrent Neural Networks with Skip and Feedback Connections.** Similar to "Spatial-Temporal" family, the input is first split and reshaped into 22 vectors of size $5 \times 7 \times 90$. And then the vectors are fed into the network one

by one in order of time. The structures of networks in this family are shown in Table 3 with additional edges. The "RNN_lstm" and "RNN_gru" is just adding LSTM/GRU between fc8 and fc_combine9 with number hidden units being 512.

## 2.3 Visualization related

**Calculating category and object level RDMs.** After the network training finished, we took the network with best performances in each family and generated the responses of each layer on validation set. For category RDM, we calculated the category mean of responses for each layer and then calculate the RDM based on that using Pearson correlation coefficient between each category. For object level RDM, the average was computed for each object for each layer and then the RDM was calculated from that using Pearson correlation coefficient between each object.

**Object grouping.** The categories are regrouped using category level RDM to show that the categorization is reasonable. The group is shown in the confusion matrix visualization in main paper, with 117 categories into 10 groups. Specifically, we took the RDM of the highest hidden layer in our best network ("TS") and used affinity propagation to group them [4] with parameter "damping" being 0.9. The results are shown in Table 4.

**MDS embedding of RDMs.** In order to estimate variance caused by initialization of the DNNs, we trained 5 networks for both "TS" model and "RNN_fdb" model from different initializations. We then generated the responses of these 10 models on validation set and computed object level RDMs based on the responses. For "RNN_fdb", we take the responses of last time point of each layer to compute the RDMs. We also tried RDMs computed from averages and concatenations of responses from last half of time points, the results are similar.

After calculating RDMs, we computed Pearson correlation coefficient between RDMs to get a distance matrix between them. Then we used MDS algorithm for to get a 2D embedding [1].

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

Table 1: Network structure for S family

| Name | Structure |
|---|---|
| S | conv1($9 \times 3$, 96), pool1($3 \times 1$, $3 \times 1$), conv2($3 \times 3$, 128), pool2($3 \times 3$, $2 \times 2$), conv3($3 \times 3$, 192), conv4($3 \times 3$, 192), conv5($3 \times 3$, 128), pool5($3 \times 3$, $2 \times 2$), fc6(2048), fc7(1024), fc_combine8(117) |
| S_more | conv1($9 \times 3$, 96), pool1($3 \times 1$, $3 \times 1$), conv2($3 \times 3$, 256), pool2($3 \times 3$, $2 \times 2$), conv3($3 \times 3$, 384), conv4($3 \times 3$, 384), conv5($3 \times 3$, 256), pool5($3 \times 3$, $2 \times 2$), fc6(4096), fc7(1024), fc_combine8(117) |
| S_few | conv1($9 \times 3$, 32), pool1($3 \times 1$, $3 \times 1$), conv2($3 \times 3$, 64), pool2($3 \times 3$, $2 \times 2$), conv3($3 \times 3$, 96), conv4($3 \times 3$, 96), conv5($3 \times 3$, 64), pool5($3 \times 3$, $2 \times 2$), fc6(256), fc7(128), fc_combine8(117) |
| S_4c2f | conv1($9 \times 3$, 96), pool1($3 \times 1$, $3 \times 1$), conv2($3 \times 3$, 128), pool2($3 \times 3$, $2 \times 2$), conv3($3 \times 3$, 192), conv4($3 \times 3$, 128), pool4($3 \times 3$, $2 \times 2$), fc5(2048), fc6(1024), fc_combine7(117) |
| S_3c2f | conv1($9 \times 3$, 96), pool1($3 \times 1$, $3 \times 1$), conv2($3 \times 3$, 388), pool2($3 \times 3$, $2 \times 2$), conv3($3 \times 3$, 128), pool3($3 \times 3$, $2 \times 2$), fc4(2048), fc5(1024), fc_combine6(117) |
| S_2c2f | conv1($9 \times 3$, 96), pool1($3 \times 1$, $3 \times 1$), conv2($3 \times 3$, 144), pool2($6 \times 6$, $4 \times 4$), fc3(2048), fc4(1024), fc_combine5(117) |
| S_1c2f | conv1($9 \times 3$, 96), pool1($6 \times 3$, $6 \times 2$), fc2(896), fc3(512), fc_combine4(117) |
| S_3c1f | conv1($9 \times 3$, 96), pool1($3 \times 1$, $3 \times 1$), conv2($3 \times 3$, 128), pool2($3 \times 3$, $2 \times 2$), conv3($3 \times 3$, 192), pool3($3 \times 3$, $2 \times 2$), fc4(1532), fc_combine5(117) |
| S_2c1f | conv1($9 \times 3$, 96), pool1($3 \times 1$, $3 \times 1$), conv2($3 \times 3$, 128), pool2($3 \times 3$, $2 \times 2$), fc3(708), fc_combine4(117) |
| S_3c0f | conv1($9 \times 3$, 72), pool1($3 \times 1$, $3 \times 1$), conv2($3 \times 3$, 144), pool2($3 \times 3$, $2 \times 2$), fc_combine3(117) |
| S_2c0f | conv1($9 \times 3$, 72), pool1($3 \times 1$, $3 \times 1$), fc_combine2(117) |
| S_1c0f | fc_combine1(117) |
| S_3D | conv1($9 \times 2 \times 2$, 96), pool1($4 \times 1 \times 1$, $4 \times 1 \times 1$), conv2($3 \times 2 \times 2$, 256), pool2($3 \times 1 \times 1$, $2 \times 1 \times 1$), conv3($3 \times 2 \times 2$, 384), conv4($3 \times 2 \times 2$, 384), conv5($3 \times 2 \times 2$, 256), pool5($3 \times 3 \times 3$, $2 \times 2 \times 2$), fc6(4096), fc7(1024), fc_combine8(117) |
| S_deep | conv1($5 \times 3$, 64), conv2($5 \times 3$, 64), pool2($3 \times 1$, $3 \times 1$), conv3($2 \times 2$, 128), conv4($2 \times 2$, 128), pool4($3 \times 3$, $2 \times 2$), conv5($3 \times 3$, 192), conv6($3 \times 3$, 192), conv7($3 \times 3$, 192), conv8($3 \times 3$, 192), conv9($3 \times 3$, 128), pool9($3 \times 3$, $2 \times 2$), fc10(2048), fc11(1024), fc12(512), fc_combine13(512), fc_combine14(117) |

Table 2: Network structure for "Temporal-Spatial" and "Spatial-Temporal" family

| Name | Structure |
|---|---|
| TS | conv1(9, 64), pool1(3, 3), conv2(3, 256), pool2(3, 2), conv3(3, 384), conv4(3, 384), conv5(3, 256), pool5(3, 2), spatial regroup, conv6($1 \times 1$, 896), conv7($1 \times 1$, 512), conv8($3 \times 3$, 384), conv9($3 \times 3$, 384), conv10($3 \times 3$, 256), fc11(2048), fc12(512), fc_combine13(512), fc_combine14(117) |
| TS_few | conv1(9, 64), pool1(3, 3), conv2(3, 192), pool2(3, 2), conv3(3, 256), conv4(3, 256), conv5(3, 192), pool5(3, 2), spatial regroup, conv6($1 \times 1$, 512), conv7($1 \times 1$, 384), conv8($3 \times 3$, 256), conv9($3 \times 3$, 256), conv10($3 \times 3$, 192), fc11(1024), fc12(512), fc_combine13(512), fc_combine14(117) |
| ST | conv1($2 \times 2$, 256), conv2($2 \times 2$, 384), conv3($2 \times 2$, 512), conv4($2 \times 2$, 512), conv5($2 \times 2$, 512), conv6($2 \times 2$, 384), temporal concatenating, conv7(1, 1024), conv8(1, 512), conv9(3, 512), conv10(3, 512), conv11(3, 512), conv12(3, 512), pool12(3, 2), fc13(1024), fc_combine14(512), fc_combine15(117) |

Table 3: Network structure for Recurrent Neural Networks family

| Name | Structure | Additional Edges |
|---|---|---|
| RNN | conv1($2 \times 2$, 96), conv2($2 \times 2$, 256), conv3($2 \times 2$, 384), conv4($2 \times 2$, 384), conv5($2 \times 2$, 384), conv6($2 \times 2$, 256), fc7(2048), fc8(1024), fc_combine9(1024), fc_combine10(512), fc_combine11(117) | |
| RNN_byp | conv1($2 \times 2$, 96), conv2($2 \times 2$, 128), conv3($2 \times 2$, 256), conv4($2 \times 2$, 384), conv5($2 \times 2$, 384), conv6($2 \times 2$, 256), fc7(1024), fc8(512), fc_combine9(1024), fc_combine10(512), fc_combine11(117) | conv1→conv3, conv1→conv4, conv2→conv4, conv2→conv5, conv3→conv5, conv3→conv6, conv4→conv6, conv2→fc7, conv4→fc7 |
| RNN_fdb | conv1($2 \times 2$, 96), conv2($2 \times 2$, 128), conv3($2 \times 2$, 256), conv4($2 \times 2$, 384), conv5($2 \times 2$, 384), conv6($2 \times 2$, 256), fc7(1024), fc8(512), fc_combine9(1024), fc_combine10(512), fc_combine11(117) | conv3→conv2, conv4→conv2, conv4→conv3, conv5→conv3, conv5→conv4, conv6→conv4, conv6→conv5, conv2→fc7, conv4→fc7 |

Table 4: Results for category regroup based on RDM

| Group | Categories |
|---|---|
| 1 | table-tennis table, folding chair, grand piano, lawn chair, rocking chair, windsor chair, swivel chair, park bench, armchair, straight chair, chair |
| 2 | drafting table, kitchen table, secretary, pool table, piano, berth, worktable, console table, easy chair, laptop, bench, coffee table, desk, table |
| 3 | upright, basket, birdhouse, platform bed, vertical file, dishwasher, convertible, monitor, love seat, printer, microwave, washer, file, stove, bookshelf, dresser, tweeter, bathtub, loudspeaker, cabinet, sofa |
| 4 | camera, school bus, bag, pendulum clock, mailbox, planter, bus |
| 5 | floor lamp, dagger, delta wing, revolver, propeller plane, carbine, knife, sniper rifle, guitar, rifle, airplane, jet |
| 6 | ferry, cabin cruiser, sea boat, cruise ship, yacht, wheeled vehicle, pistol, ship, tender, train, boat |
| 7 | limousine, ambulance, stock car, roadster, jeep, beach wagon, wine bottle, cruiser, sports car, convertible, bottle, racer, sport utility, sedan, coupe, car |
| 8 | cap, soda can, coffee mug, jar, mug, helmet, bowl, pot, ashcan, vase |
| 9 | data input device, remote control, pillow, telephone, screen, clock, liquid crystal display, cellular telephone |
| 10 | microphone, earphone, sailboat, motorcycle, table lamp, faucet, lamp |