[Reviews · NeurIPS 2017]

Reviewer 1



The study represents a systematic approach to simulation and modeling of the rodent vibrissal system all the way from sensory transduction to behavior (discrimination of object classes). Thanks to the high ethological relevance and the richness of existing empirical knowledge about the coding of information by the vibrissal system, this direction of work will likely yield new insights about the principles of sensory coding, generate new hypotheses for empirical studies, and allow capturing such principles in artificial systems and algorithms. The comprehensive nature of the approach will allow iterative refinement and validation of each aspect of the model in one framework. As the authors indicate, the present study is a proof-of-concept investigation paving the way for future studies to establish close correspondence between biological and artificial sensory systems. Their framework will allow future projects to incorporate more realistic empirically derived physical models of the whiskers, the empirically measured properties of various mechanoreceptor types, the known functional differences of the distinct pathways with their respective sensory transformations, integration of active whisking, etc. Since this is a proof-of-concept study, it seems inappropriate to criticize the study for not integrating many of the known properties of the vibrissal system (e.g. characteristics of the mechanoreceptors) -- these will be integrated in future iterations. The networks were trained used a supervised approach. It would be great to see a discussion of the biological counterparts of the error signals. I disagree with the authors' assessment that models with vastly different numbers of parameters should not be directly compared. This concern only holds when training and testing on the same data, giving an unfair advantage to models with more degrees of freedom. However, with proper cross-validation and testing procedures prevent such mis-characterizations. Furthermore, regularization schemes complicate the estimation of the true number of free parameters in the models. As the authors suggest, the use DRMs indeed may hold the promise of matching brain areas or layers with corresponding level of representation. Overall, I find that the study outlines several directions for further investigation in both computational and empirical studies and I recommend accepting the paper.

Reviewer 2



The authors develop several models of the Whisker-Trigeminal system, each with a different neural architecture, trained to solve a shape-recognition task. The models are composed of a module capturing simplified but biologically-motivated whiskers physics, dynamics and transduction, and a deep neural network loosely inspired in the hierarchical structure of the first stages of the somatosensory system. Several neural network architectures are compared in terms of task performance, and a few of those achieve reasonable performance. Overall, the paper is technically sound, and is clearly written, with the methods and assumptions clearly exposed. However, the paper would have gained if a tighter link would have been established between the models and the specific neural areas, and if more concrete experimental predictions would have been provided: -in lines 255-258, the authors write “The key biologically relevant follow-up question is then: how should we distinguish between the elements in the shortlist?”. Arguably, as important as this question is to ask what the successful models have in common. Indeed, the authors list the architectural features that seem to correlate with good performance (lines 235-251), but it would be important to also discuss how these features map into the neural system and could be tested experimentally; -the proposal to experimentally validate the different network architectures by comparing the Representational Dissimilarity Matrices in models and neural recordings seems reasonable but more details should be provided: what are the specific predictions for the different models? Also, could a quick analysis be done on publicly available experimental datasets to test which architecture leads to the closest results? Some typos: -Figure 1 caption, second to last line, instead of “then measure to the”, “then measure the”; -Figure 2 caption, instead of “to to the follicle”, “to the follicle”; -Figure 2 caption, instead of “variable axes is been”, “variable axes has been”; -line 73, instead of “to the reasonably”, “to be reasonably”; -line 138, instead of “implementary”, “supplementary”; -line 267, instead of “in general are comparatively”, “in general comparatively”; -line 320, instead of “might would be”, “might be”. -- after the rebuttal -- I read the rebuttal and my colleagues reviews, and would like to raise the score from 6 to 7, therefore recommending acceptance.

Reviewer 3



* Summary The authors of the paper "Toward Goal-Driven Neural Network Models for the Rodent Whisker-Trigeminal System" train networks for object detection on simulated whisker data. They compare several architectures of DNNs and RNNs on that task and report on elements that are crucial to get good performance. Finally they use a representational dissimilarity matrix analysis to distinguish different high performing network architectures, reasoning that this kind of analysis could also be used on neural data. * Comments What I like most about the paper is direction that it is taking. Visual datasets and deep neural networks have shown that there are interesting links between artificial networks and neuroscience that are worthwhile to explore. Carrying these approaches to other modalities is a laudable effort, in particular as interesting benchmarks often have advanced the field (e.g. imagenet). The paper is generally well written, and the main concepts are intuitively clear. More details are provided in the supplementary material, and the authors promise to share the code and the dataset with the community. I cannot judge the physics behind simulating whiskers, but the authors seem to have taken great care to come up with a realistic and simulation, and verify that the data actually contains information about the stimulus. I think a bit more nuisance variables like slight airflow, as well as active whisking bouts would definitely make the dataset even more interesting, but I can see why it is a good idea to start with the "static" case first. With regard to benchmarking, the authors made an effort to evaluate a variety of different networks and find interesting insights into which model components seem to matter. I would have wished for more details on how the networks are setup and trained. For example, how many hyperparameters were tried, what was the training schedule, was any regularization used, etc? The code will certainly contain these details, but a summary would have been good. The part about the representation dissimilarity matrix analysis is a bit short and it is not clear to me what can be learned from it. I guess the authors want to demonstrate that there are detectable differences in this analysis that could be used to compare neural performance to those networks. The real test for this analysis is neural data, of course, which the paper does not cover. In general, I see the main contribution of the paper to be the dataset and the benchmarking analysis on it. Both together are a useful contribution to advance the field of neuroscience and machine learning.